# Navigating the Hurdles of Intra-Articular AAV Gene Therapy

**DOI:** 10.3390/ijms262010123

**Published:** 2025-10-17

**Authors:** Wenjun Li, Owen Thornton, Susi Feng, Chengwen Li

**Affiliations:** 1Gene Therapy Center, University of North Carolina at Chapel Hill, Chapel Hill, NC 27599, USA; tia119@unc.edu (W.L.); susi_feng@med.unc.edu (S.F.); 2Division of Oral and Craniofacial Biomedicine, University of North Carolina Adams School of Dentistry, Chapel Hill, NC 27599, USA; 3Department of Psychology and Neuroscience, University of North Carolina at Chapel Hill, Chapel Hill, NC 27599, USA; othornt@unc.edu; 4Department of Pediatrics, University of North Carolina at Chapel Hill, Chapel Hill, NC 27599, USA; 5Carolina Institute for Developmental Disabilities, University of North Carolina at Chapel Hill, Chapel Hill, NC 27510, USA

**Keywords:** adeno-associated virus, joint, barriers, AAV engineering, immune management

## Abstract

Joint diseases represent a significant health burden due to their high prevalence and morbidity, yet current treatments fail to provide comprehensive and long-term relief for all patients. In this context, adeno-associated virus (AAV) gene therapy has emerged as a promising approach, offering advantages such as prolonged efficacy and minimal immunogenicity. AAV has been extensively studied for various medical conditions, with some applications successfully implemented in patient treatments. Currently, a few clinical trials utilizing AAV have been completed for treating arthritis. However, challenges such as transduction efficiency, off-targets, and preexisting immune responses persist. This review provides an overview of the current paradigms of treatment with regard to joint diseases, elaborates on the AAV delivery barriers related to application in treating joint diseases, and discusses strategies to improve gene therapy efficacy, including AAV capsid engineering, small molecule-assisted AAV delivery, optimizing tissue-specific or inflammation-inducible promoters, as well as strategies to mitigate immune responses to AAV.

## 1. Introduction

Joint disease is one of the leading causes of disability. The term covers many conditions, including inflammatory and degenerative disorders, traumatic injuries, and tumors that arise in bone, muscle, tendons, ligaments, cartilage, the synovium, and the joint space. Its burden is not limited to movement or range of motion; it significantly restricts patients’ daily activities and diminishes their quality of life [1]. Additionally, these patients are usually susceptible to various psychopathologies such as depression and anxiety. Due to the considerable societal and economic impacts, constant improvements in treatment regimens are essential to relieve the current disease burden.

Patients with early to moderate symptoms are commonly managed with pharmaceutical therapy. In recent years, a subclass of disease-modifying antirheumatic drugs, known as biologics, has emerged as a promising alternative to conventional agents for joint disease, especially autoimmune forms of arthritis such as rheumatoid and psoriatic arthritis. These drugs target specific inflammatory pathways to interrupt the mechanisms that drive joint damage [2].

Despite this progress, results are often uneven. Clinical benefit may take several weeks to appear, and repeated dosing is usually required because protein drugs clear quickly from the joint space and have a short half-life in that environment [3]. The treatments typically yield a response in less than 50% of patients, and at least 10% of patients still end up with irreversible severe disability [4].

Gene therapy is a flexible platform for treating disease. It works in three principal modes: gene addition, gene silencing, and gene editing. Delivery systems fall into two broad classes: viral and nonviral vectors. Among the viral vectors, adeno-associated virus (AAV) stands out because it is considered nonpathogenic, supports durable gene expression, and can be adapted to many clinical indications.

This review broadly overviews current approaches to the treatment of joint disease, explains how they potentially benefit from using AAV as an alternative delivery vehicle, and examines the specific barriers that AAV faces in the effective, locally applied treatment of joint diseases, as well as strategies to overcome these barriers.

## 2. AAV Biology

Unlike other vectors, AAV is episomal, minimizing the risk of insertional mutagenesis, and adverse immune responses. With broad tropism and the ability to efficiently transduce both dividing and non-dividing cells, different AAV serotypes can be utilized to target specific tissues, like the brain, liver, and muscles, making it versatile for treating various diseases. Most importantly, one time administration of an AAV vector is able to induce long-term expression of the packaged therapeutic genes, making it an ideal vector for the treatment of joint diseases.

AAV was first identified in 1965 as an unexpected contaminant in a simian adenovirus preparation. In the early 1980s, it was shown to be replication-defective and nonpathogenic, making it an appealing candidate for gene therapy [5]. The earliest clinical trial using AAV began in 1993 for cystic fibrosis. Expression levels were low, and clinical benefit was limited at that time, yet the study accelerated interest and development of AAV vectors.

The key difference between wild-type AAV and recombinant AAV (rAAV) is their genetic composition. Both retain identical inverted terminal repeats (ITRs), which are essential for genome replication, encapsidation, transcription, and integration [6]. In rAAV, the transgene and its promoter occupy no more than 4.7 kb between the ITRs, replacing the rep and cap genes that fill that space in the wild-type virus.

rAAV transduction [7] begins when the capsid binds to cell surface receptors and coreceptors, then enters the cell through endocytosis. Inside acidic endosomes, conformational shifts expose the N termini of VP1 and VP2, which facilitates particle escape and movement toward the perinuclear region. A portion of virions undergo protease-mediated degradation, while the rest traffic to the nucleus. After nuclear entry, the capsid uncoats and releases the rAAV genome. The single-stranded DNA is converted by host enzymes into a double-stranded form. The vector genome usually persists as episomes, often circularized by DNA repair pathways, and less often integrates into the host genome. The DNA is then transcribed, leading to translation of the encoded therapeutic gene.

## 3. Clinical Trials

AAV vectors have advanced through many clinical trials, and several have earned FDA and EU approvals for inherited retinal dystrophy, spinal muscular atrophy, and hemophilia [8]. In 2017, Luxturna (voretigene neparvovec), which uses an AAV2 vector, became the first FDA-approved AAV therapy for an inherited disorder, treating RPE65-related retinal disease. Zolgensma (onasemnogene abeparvovec-xioi) followed in 2019, delivered a functional SMN1 gene to motor neurons with an AAV9 vector for patients with spinal muscular atrophy (SMA). Hemgenix received approval in 2022 for hemophilia B, and Roctavian was approved in 2023 for hereditary hemophilia A. In Europe, Upstaza won approval for aromatic L-amino acid decarboxylase deficiency (AADC), and Roctavian was approved for hemophilia A.

Within joint gene therapy, the story is still in its early chapters. Preclinical studies using AAV for arthritis have shown promise, yet no AAV-based therapy has been approved for arthritis in patients to date. Currently, the two major routes for gene therapy administration in joint diseases are systemic and intra-articular injections [9]. Joint defects are good candidates for intra-articular injections because it is possible to treat the joint independently without impacting other joints, if not damaged, and only a negligible amount of AAV will circulate in the blood and liver due to the low virus titer required for intra-articular injection relative to other injection routes. Compared with traditional drugs, which struggle to maintain steady pharmacokinetics and stable protein or chemical concentrations, AAV vectors can transduce joint cells and provide long-term potency.

By searching the website of clinicaltrial.gov, currently, there are only seven clinical trials for joint disease that are established or ongoing (Table 1) [10], mostly targeting a single cytokine, such as IL-1Ra and TNFα [11,12]. Since only a small number of organizations have conducted clinical trials to date, with some sponsors running two or more studies, the serotype selections remain limited. Most clinical reports have focused on AAV2 and AAV5, largely because these vectors were the earliest to demonstrate efficiency in musculoskeletal tissues and had already accumulated substantial preclinical experience, and because those serotypes transduce synovial fibroblasts and chondrocytes. However, they are not the only serotypes with potential. Studies have also shown that AAV1 and AAV6 achieve strong transduction efficiency in joint tissues [10], suggesting there may be more effective candidates for future clinical translation.

One such trial is a phase I/II clinical trial of an AAV vector encoding the interleukin-1 receptor antagonist (IL-1Ra) for the treatment of knee osteoarthritis (OA). IL-1Ra is a naturally occurring protein antagonist of interleukin-1, an inflammatory cytokine tied to the development and progression of OA [13]. In a knee OA study, investigators assessed the safety and tolerability of IL-1Ra gene therapy and measured its effects on pain and function. Two additional clinical trials in OA evaluated ICM-203, an AAV vector encoding Nkx3.2, a transcription factor involved in the activity of joint cells such as chondrocytes and synoviocytes. Based on an interim review, in general, some patients experienced improvements in metrics such as pain, synovitis, and functionality, but with mixed results, with further investigations into higher doses likely necessary for more conclusive findings. For rheumatoid arthritis (RA), two phase I trials used AAV vectors encoding soluble tumor necrosis factor receptor 1 (sTNFR1). Two others encoded hIFN-β. In the hIFN-β studies, a single intra-articular dose of recombinant AAV type 2/5 carrying the hIFN-β gene was delivered into the carpometacarpal (CMC), metacarpophalangeal (MCP), proximal interphalangeal (PIP), distal interphalangeal (DIP), or wrist joint [8]. However, this clinical trial using hIFN-β was suspended due to issues with tolerability; specifically, it was reported that several patients experienced long-lasting local adverse events, including joint swelling, periarthritis, tenosynovitis, and pain. Though no severe systemic side effects were observed, the persistent local response led to termination of further enrollment [14]. Overall, the limited benefits observed in these clinical trials indicate the need to explore alternative targets.

## 4. Preclinical Targets

Joint diseases usually present with pain, stiffness, and swelling. To manage current disease and build better options, many preclinical studies have mapped promising targets. Conditions such as RA, psoriatic arthritis, SLE, gout, and AS share chronic inflammation or other immune dysfunction. The mechanism of the inflammation process is a complex network—dysregulated immune cells interact with cytokines, chemokines, enzymes, and antibodies against a background of genetic variation.

One major strategy is to reset abnormal signaling pathways. Delivery of immunomodulatory cytokine genes with AAV, including IL-4, IL-10 [15,16], IFN-β [17], and TGF-β, PD-L1 [18], has been reported to reduce joint inflammation, ease pain, and help prevent structural damage in animal models. These anti-inflammatory cytokines act in part by shifting immune cell profiles: they promote the expansion of regulatory T cells (Treg) and M2-polarized macrophages, while reducing pro-inflammatory M1 macrophages, thereby suppressing synovial inflammation and restoring immune balance. Specifically, IL-10 and IL-4 may be prioritized for conditions like RA where inflammation predominates, whereas TGF-β may be more valuable in OA, where cartilage degeneration are the main drivers [15,16]. A complementary approach blocks pro-inflammatory cytokines. Antagonists of IL-1 [19], IL-6 [20], and IL-17 [21] limit bone destruction and autoimmunity [22]. Lastly, signal transduction pathways, including NF-κB, MAPKs, and JAK-STAT pathway, have also been implicated as potential inhibition targets [23]. Synergistic strategies can be considered, for instance, combining antagonists of IL-1 and IL-10 delivery to simultaneously dampen inflammation and promote tolerogenic immune pathways. However, such approaches carry an increased risk of excessive immunosuppression, which could predispose patients to infection or even malignancy.

Another major approach to treating joint disease is dampening the effect of dysfunctional immune cells, which play a direct role in the manifestation of physical symptoms; for instance, cartilage degradation and synovial fibroblast proliferation in RA are largely induced by the signaling of pro-inflammatory cytokines released by T cells and other white blood cells, including B cells and neutrophils. In particular, the delivery of checkpoint proteins such as cytotoxic T-lymphocyte-associated protein 4 (CTLA4) and Programmed death-ligand 1 (PD-L1) have been indicated in multiple autoimmune diseases as a method of reducing dysfunctional T-cell activity [18]. Previously, we demonstrated the potential of a soluble PD-L1 protein, delivered and expressed by AAV6 intra-articularly, to attenuate collagen-induced arthritis (CIA) severity in a mouse model in vivo.

A direct way to alter the immune system is to block the receptors that those cells display on their surface. Antibodies or inhibitors can do this specifically with a cell-type focus, including agents against CD19 or CD20 to deplete B cells; CD3 or CD4 to modulate T-cell activity [24]; CD64 or GM-CSF to drive macrophage repolarization toward a reparative profile [25]; and CR3 or DEC-205 to target dendritic cells [26].

Besides immunological targets, research can also focus on cartilage regeneration, angiogenesis, and pain relief. For example, while AAV-delivered matrix metalloproteinases (MMPs) and their inhibitors (TIMPs) have been applied to treat glaucoma and tumors [27], overexpression of MMPs is also associated with cartilage degradation, and serves as a potential indicator of the onset of OA and RA [28]. In addition, various growth factors have also demonstrated promise in chondral and vascular regeneration [29,30]; for example, AAV-delivered TGF-β was proved to efficiently repair cartilage explants [29].

Overall, rational target selection requires balancing joint disease subtype, inflammatory stage, and cell-type specificity to optimize both efficacy and safety.

## 5. Barriers and Solutions in AAV Gene Therapy

While AAV gene therapy shares common complications such as limited cargo size, toxicity, and immunogenicity, joint delivery adds three central hurdles (Table 2): first, sufficient transduction efficiency in the joint; second, prevention of vector escape into the bloodstream, where off-target transduction in unrelated tissues can lead to complications; and third, overcoming preexisting neutralizing antibodies in synovial fluid, since Nabs to AAV are common in human populations and can largely compromise therapeutic effects.

### 5.1. Transduction Efficiency

Therapeutic success rises and falls with gene transduction in the target tissue and cells. With systemic routes such as intravenous injections, a large share of expression ends up in the liver, which drains signal away from the joint and leaves much lower intra-articular expression. When local injections are administered, more transgene expressions will be located at the injected sites, which are typically the primary areas of symptoms. However, there is still plenty of room to improve the general transduction efficiency. Specifically, innovations in AAV technology can be used to address three main issues with general transduction efficiency: first, the range of cell types being transduced is limited, as certain cells, such as lymphocytes and myogenic stem cells, are difficult to transduce; second, the percentage of transduced cells typically reaches only 20–30% [18], leaving substantial potential for increasing protein expression; and lastly, the amount of protein produced per transduced cell unit can also be enhanced.


Potential Strategies:

Although early clinical programs relied on wild-type AAV serotypes, experience has revealed constraints inherent to native capsids, including limited tissue tropism, suboptimal transduction at target sites, and clinically relevant immunogenicity. Capsid engineering offers a systematic path to address these liabilities and to produce vectors with improved cell selectivity, greater efficiency, and a lower risk of immunogenicity.

Rational design strategies use peptide insertion, loop grafting, and domain exchange to create capsids with new properties. Exchanging variable regions between serotypes can combine favorable features from each parent while avoiding undesirable traits. The resulting vectors can display refined tropism and higher transduction efficiency that supports intra-articular delivery. Activity-dependent designs further illustrate the promise of this approach. In a provector configuration, insertion of a matrix metalloproteinase cleavage motif renders the particle protease-activated within MMP-rich microenvironments, a hallmark of many inflammatory conditions, including arthritis [31]. This yields spatial and temporal control of transduction, restricts expression to inflamed tissue, and limits exposure in non-target sites. Taken together, these strategies point toward AAV vectors that achieve therapeutic effect at lower doses with fewer immune sequelae, which is the profile most compatible with joint disease therapy.

Even small changes to the capsid can yield significant effects on transduction efficacy. Single amino acid alterations can impact the capsid structure for receptor binding and antibody escape, as well as hydrogen bonding, which changes the stability of the capsid, or post-translational modification processes, including acetylation, phosphorylation, ubiquitination, SUMOylation, and so on [32], which decrease the proteasome-mediated degradation and enhance trafficking to Golgi [33]. For instance, previous studies showed that deletion of threonine in VRI increased transduction efficiency and antibody escape in mouse joints [34].

Next, directed evolution can be used to isolate promising candidates from many random variants. This could be explored by screening large mutant capsid libraries through random mutagenesis of the capsid, via DNA shuffling or high-throughput approaches [35] and then selecting for variants with desired properties, such as specific tissue, cell type, or enhanced transduction. A study [35] created a library with cap genes from parent AAV serotypes and then ran error-prone PCR to generate random cap chimeras. The virus, after a few cycles of selection to identify the highest viral titer and further verify other properties, was then cultured in specific cell plates relevant to the target cell types. The selection can also be performed by in vivo screening, for example, by testing which mutated variant was mostly found in AAV-injected mice or primates [36]. This method requires a large mutant library and significant labor if fully relied upon. Nevertheless, with advancements in bioinformatics technology, the efficiency of the process has been greatly accelerated, leading to the identification of a few promising candidates from thousands of options.

AAV genome engineering includes modifying the transgene cassette that will be encapsulated in the AAV capsid. It involves codon optimization [37] or self-complementary ITRs to enhance transduction in certain cells [38]. ITR engineering can improve multiple attributes of AAV vectors. A good example is the self-complementary AAV [39]. This design is created by mutating one terminal resolution site within an ITR so the essential single-strand nick cannot occur [39]. The packaged genome becomes a dimeric inverted repeat that folds into a duplex after uncoating. Because the vector now bypasses second-trand synthesis, the onset of transgene expression is shortened. Altering CpG content in the ITR provides another lever. Unmethylated CpG motifs can trigger strong innate immune activation, so CpG-free ITRs have been developed to attenuate these responses [40]. ITR modification may also support dual-vector strategies by promoting heterodimer formation and episomally stable concatemers. Gene expression from current dual-delivery systems remains limited, since only heterodimers that form in the same orientation can be recombined into a full-length transgene [41].

Other elements in the process of transgene expression, including introns, post-transcriptional regulatory elements, polyadenylation signal sequences, enhancers, and so on, are also related to the therapeutic efficiency of AAV-delivered material and have been adopted as targets for optimization. For instance, incorporating scaffold or matrix attachment elements into AAV genomes can improve efficiency. These A/T-rich DNA sequences tether plasmid or vector DNA to the nuclear matrix and have been reported to keep AAV concatemers in a more open chromatin state, which supports durable transgene expression [42]. However, the issues of increased vector size and unintended epigenetic change on host genome, among others, increase the risk associated with using this system.

Small molecules and biomaterials can also raise the ceiling on trafficking, uncoating, and nuclear entry. Proteasome inhibitors such as bortezomib limit degradation of ubiquitinated capsids and thereby increase AAV transduction. Histone deacetylase inhibitors, including FK228 [43] and depsipeptide [44], have been shown to raise transduction, associated with the accumulation of acetylated histones that favor transcription of the delivered cassette. In one study, investigators mixed AAV with a three percent alginate solution in PBS and crosslinked the gel in calcium chloride; alginate-mediated AAV delivery of IGF I supported the repair of a chondral defect [45]. Genome engineering can further boost vector performance by altering host DNA repair. Engineered nucleases such as zinc finger nucleases (ZFNs) and transcription activator-like effector nucleases (TALENs) can induce double-stranded breaks in the cellular genome, which promotes uptake and recombination of AAV-delivered sequences [46]. This strategy has been reported to enhance gene targeting in cells by about twenty-five percent [35]. However, creating targeted breaks can threaten genome stability, so any gain in efficiency must be weighed against the risk profile for the intended indication.

### 5.2. AAV Toxicity and Safety Considerations

Potential risks after AAV administration include unintended gene expression in non-target tissues and more severe injury, including malignancy, acute inflammatory reactions, or even fatal outcomes. Multiple factors influence AAV toxicity. These include overexpression of the transgene or excessive capsid load, manufacturing problems such as insufficient purification or contamination with wild-type AAV, high doses, and the influence of host immunity and genetic variation.

#### 5.2.1. Integration

Integration is less common with AAV than with lentiviral vectors, yet it is not rare. Canonical integration of wild-type AAV occurs at chromosome 19 and depends on the cis and trans functions of REP78 and REP68 and their binding sequences. Recombinant vectors used clinically do not supply REP, and most vector genomes persist as episomes, which can support long-term expression, although integration events are still detected [47,48]. The safety concern is insertional mutagenesis, which could activate oncogenes or inactivate tumor suppressor genes. To date, across more than two hundred clinical studies, there is no direct evidence that recombinant AAV has caused tumor formation in humans [49]. The data from animal experiments are mixed. Russell et al. reported a higher incidence of hepatocellular carcinoma in normal newborn mice after intravenous AAV [50], whereas other investigations have shown a more favorable profile. In a long-term follow-up of twelve macaques that received liver-directed AAV, Greig et al. [47] found no marked clonal expansion linked to cancer fifteen years after treatment.

#### 5.2.2. Off Target

Off-target events are relevant to route, dose, and serotype. Systemic administration carries the greatest risk, since intravenous dosing sends a large fraction of expression to the liver, which is the dominant off-target organ. However, studies aimed at liver correction have shown transgene mRNA confined to the liver without chronic liver disease or malignancy, with durable correction that can last for decades in that setting [51]. With intra-articular injection, some serotypes remain essentially confined to the injected joint, whereas others cross into the circulation and reappear in the liver. In one comparison across AAV1 through AAV9, vectors 1 to 6 were detected only in the injected joint, while AAV7, AAV8, and AAV9 produced strong expression in both the joint and the liver [34]. Therefore, even with local dosing, very high vector loads can potentially travel beyond the target, depending on the capsid characteristics.

#### 5.2.3. Immunogenicity

Immune responses to the capsid or to the transgene vary by serotype, by species, and by the transgene itself [52]. Factors that shape immunity include vector dose, CpG content in the genome, biochemical features of the expressed protein, host immune status, route of administration, age, and patient HLA type [53]. Recent experience with very high doses of near 2 × 10^14^ vector genomes per kilogram in nonhuman primates revealed severe hepatotoxicity and pathology in the dorsal root ganglia [54].

Innate recognition follows several pathways. TLR2 can engage when the capsid binds at the cell surface, and TLR9 senses unmethylated CpG during endosomal trafficking. Cytosolic DNA sensing through cGAS and STING, and RNA sensing through RIG I and MDA5, add further inputs. AIM2 and IFI16 can promote inflammasome activation. Activated dendritic cells then further activate CD8 and CD4 lymphocytes, which recognize peptides presented on MHC I and MHC II and can eliminate AAV-transduced cells [55].

Transgene products also elicit a range of immune responses. Both secreted and intracellular proteins can be immunogenic, with secreted proteins more likely to drive combined peripheral and central responses. Antibodies generated by B cells can neutralize the protein and recruit complement, which further assists antibodies in clearing pathogens and damaged cells. Complement engagement begins when antibody-coated particles are recognized by C1, leading to activation of C3 and C5 and formation of the membrane attack complex. The result is opsonization, phagocyte recruitment, and, in some settings, direct lysis [56].

Potential Strategies:

Promoter selection is a critical point for transgene expression. The promoter can be selected or designed to match disease according to the spatial and temporal dynamics of arthritis, where cell states change across flares and remission. Many trials use strong, ubiquitous, and high-expressing promoters such as variants of the human cytomegalovirus (CMV) or chicken beta-actin (CBA) promoters with a CMV enhancer (CAG). Nevertheless, greater precision will likely require cell-selective control, including promoters for specific chondrocyte subsets, synoviocytes, or newly emergent inflamed cell populations. Chondrocytes turn over slowly, which makes them attractive for stable AAV expression. Cartilage-specific promoters such as Col2a1, Col11a1, and aggrecan have been explored. Col2a1 drives broad expression in chondrocytes, whereas Col10a1 is active in hypertrophic chondrocytes [57]. Bennett et al. identified cartilage-specific control elements within intron 2 of the alpha 2 type I collagen gene [58]. A complementary route is genome editing that inserts the therapeutic sequence at an endogenous locus so that native regulatory elements control expression, waiving the need for an exogenous promoter [59]. Each option must be sized against the AAV size limit of approximately 4.7 kb, since many tissue-specific promoters are larger than 1 kb.

Another innovation to regulate transgene activity is designing the vector with RNA molecules. For example, microRNA (miRNA), a single-stranded noncoding RNA that binds to the target receptors or sequences, has been applied to assist AAV internalization and regulate transgene transduction in specific cell types [60]. Nevertheless, this method requires careful optimization to mitigate unintended silencing of genes and enhance vector stability.

Besides regulating genes with tissue specificity, another concern is controlling gene expression in a timely manner. Arthritis comes with flare-ups and remissions; if the treatment matches disease progression, it will largely decrease the potential side effects. Studies show that inflammation may increase transgene expression across different AAV contexts, independent of serotype and promoter selection, or single-stranded versus self-complementary design [61]. In RA, stronger and more durable AAV transduction has been observed compared with healthy joints [62]. The likely contributors are inflammation-linked changes in cellular metabolism and tissue physiology [63], including higher blood flow and greater vascular permeability at the affected site. Oxidative stress may add to this effect. Reactive oxygen species can promote viral entry by aiding endosomal escape and accelerating intracellular trafficking [64].

A plausible way to align expression with disease activity is to use inflammation-responsive promoters. Placing these elements upstream of a therapeutic cassette allows flares to raise expression while remission lowers it. The most commonly reported inflammation-responsive promoters include IL-1, IL-6, NFκB-responsive promoters [61], MMP, activator protein 1 (AP-1), complement C3–based promoter, cyclooxygenase (COX)-2–based promoter [65], and chemokine (C-X-C motif) ligand 1 (CXCL1) [66]. However, most of those studies remain in vitro across varied cell lines, and animal data are limited.

Harnessing the local immune microenvironment provides another path to decrease the immune response toward AAV. Tregs and M2-polarized macrophages are central players in maintaining immune tolerance in the joint. Expansion of Tregs can be achieved through the activation of IL-10 or TGF-β, which promotes their differentiation and suppresses effector T-cell activity. Similarly, skewing macrophages toward an M2 phenotype with cytokines such as IL-4 or IL-13 can reduce the pro-inflammatory M1 phenotype and mitigate AAV immunogenicity.

### 5.3. Preexisting Immune Response in Joint

Preexisting neutralizing antibodies (Nabs) are found in both serum and synovial fluid across many species, with prevalence shaped by species and serotype [67]. More than ninety percent of humans have been exposed to AAV, and approximately fifty percent carry neutralizing antibodies [68]. Synovial fluid, like serum, contains diverse immune cells and Nabs, but titers do not reliably mirror those in blood [69]. In OA, one study measuring anti-AAV2.5 antibodies in paired samples reported higher titers in serum than in synovial fluid [69].

T cells have also been reported to be extracted from the synovial fluid of arthritis patients, as T cells often infiltrate 30–50% of the synovial lining in RA [70,71]. However, T-cell levels also do not necessarily match Nab prevalence [72]. Anita et al. observed similar T-cell responses to multiple AAV serotypes despite differing Nab levels [72]. In healthy adults, Hildegund et al. detected AAV capsid–reactive CD8^+^ and CD4^+^ T cells in about half of individuals [73]. In pre-immunized patients, effector T cells are localized to joint tissues and can be activated once the antigen comes up again; for instance, CD8^+^ T cells will be activated into cytotoxic T lymphocytes (CTL) and induce CTL-mediated cell lysis. It was reported that severe cytotoxic T-cell response could lead to acute severe hepatitis.

Arthritic joints contain more antibodies and T cells than healthy joints. How this heightened local immunity shapes responses to AAV and to the transgene product is still debated. The immune system is skewed toward arthritis-related drivers, which may not translate into a stronger reaction against AAV and could even coincide with a weaker one, so the net effect remains uncertain. One point is clear: when patients are taking immunosuppressant drugs, overall immune activity falls, which generally helps AAV transduction.

Potential Strategies:

A strong, early immune response to AAV can clear transduced cells or generate neutralizing antibodies against the capsid or the transgene product, which shortens the effective life of therapy [74]. Clinicians may then increase the vector dose to recover expression, a tradeoff that raises the risk of toxicity and further immune activation.

To widen eligibility and extend durability, especially in seropositive patients, a brief and well-timed suppression of immunity during the AAV transduction window is key. Approaches include plasmapheresis [44]; peri-infusion immunosuppression with corticosteroids, rapamycin, anti-thymocyte globulin, CSP, MMF, or calcineurin inhibitors [75]; cleavage of circulating IgG with IdeS [76]; and decoys that mimic the antigen to soak up neutralizing antibodies [77]. The antibodies associated with disease progression could also indirectly damage tissues by the complement system. Therefore, employing complement inhibitors is another approach [78]. Several downstream inflammatory responses could also be altered through this process, such as TNF-α signaling, MyD88, IKK-γ, or NF-κB [79]. These drugs could temporarily mitigate B-cell and T-cell responses, offering opportunities for AAV to cross the vessel wall and enter cells for transduction. However, for intra-articular injection, plasmapheresis [44] or immunosuppressive drugs through blood, or other systemic alteration may not be efficient, as synovial antibody levels are not always consistent with serum levels. Direct intra-articular application of these treatments may therefore be more effective.

Careful selection of the AAV capsid, paired with lower doses, can decrease immune responses to both the capsid and the transgene. Capsid engineering, as noted above, can also reduce crosslinking triggered by preexisting antibody binding.

The delivery route is also important. Intravenous and intramuscular AAV have shown higher rates of humoral responses than intra-articular dosing, likely because distribution and clearance differ across these routes [80]. By contrast, liver targeting or oral mucosal delivery has been reported to promote immune tolerance. Studies describe dampened responses to the transgene, activation of tolerogenic DC and Treg cells, and higher levels of anti-inflammatory cytokines such as IL10 and TGF-β [81]. Isolated limb perfusion is another safe option. By confining the vector locally, it boosts transduction at lower total doses and has been associated with relatively mild T-cell responses [82].

## 6. Future Perspectives and Conclusions

While gene therapy has been applied in quite a few clinical trials and a few products are now on the market, the application of AAV to nonlethal or complex multigenic disorders like arthritis remains difficult. The sticking points are well known, including off-target effects, immunogenicity, and toxicity. Local delivery usually carries less risk than systemic delivery, which makes it the safer alternative in the current landscape of gene therapy.

In conclusion, AAV-based gene therapy is promising for joint diseases. However, challenges persist. The success of AAV clinical application depends on precise delivery of genes to the affected location during the active disease phase, limiting off-target activity and toxicity, and maintaining sustained and properly regulated expression. Future work should focus on capsids and transgene cassettes, defining the most useful targets, and reducing the AAV immune response. Finally, consideration of ethical and biosafety issues is essential for achieving greater success in joint gene therapy.

## Figures and Tables

**Table 1 ijms-26-10123-t001:** Clinical trials for AAV gene therapy in arthritis.

AAV Serotype	Gene of Interest	Indication	Location/Joints	Sponsor	Trial Number	Dose (vg)	Patients (*n*)
AAV5.2	ICM-203	OA	Knee	ICM Co., Ltd. (Tokyo, Japan)	NCT05454566	6 × 10^12^–6 × 10^13^	18
AAV5.2	ICM-203	OA	Knee	ICM Biotech Australia Pty Ltd. (Melbourne, Australia)	NCT04875754	6 × 10^12^–6 × 10^13^	16
AAV5	sTNFR1	RA	Wrist	Arthrogen (Amsterdam, The Netherlands)	NCT03445715	2.4 × 10^12^–2.4 × 10^13^	15
AAV5	sTNFR1	RA and OA	Finger joints	Arthrogen	NCT02727764	0.6 × 10^12^–1.2 × 10^13^	12
AAV2.5	IL-1Ra	OA	Knee	Mayo Clinic (Rochester, NY, USA)	NCT02790723	1 × 10^11^–1 × 10^13^	9
AAV2	TNFR	RA	Peripheral joints	Targeted Genetics Corporation (Seattle, WA, USA)	NCT00617032	1 × 10^10^–1 × 10^11^	15
AAV2	TNFR	Various arthritis	Peripheral joints	Targeted Genetics Corporation	NCT00126724	1 × 10^11^–1 × 10^13^	120

**Table 2 ijms-26-10123-t002:** Clinical barriers and potential strategies in AAV gene therapy for joint diseases.

Barriers	Description	Potential Strategies
Limited transduction efficiency	Low expression in joint tissue due to poor tropism, restricted cell range, and low percentage of transduced cells.	Capsid engineering ITR and cassette optimization Use of scaffold/matrix attachment elements Small molecules (e.g., proteasome or HDAC inhibitors) Engineered nucleases (e.g., ZFNs, TALENs)
Integration	Insertional mutagenesis.	Lower dosing with more efficient capsids Capsid engineering Tissue-specific or inflammation-responsive promoters Genome editing at safe harbor loci Monitoring vector purity to avoid contamination
Off-target distribution	Escape of AAV from joint into bloodstream and unintended transduction in liver or other tissues.	Local intra-articular administration Serotype selection Capsid engineering for restricted tropism Controlled dosing to minimize leakage
Immunogenicity	Activation of innate and adaptive immune responses to capsid and/or transgene.	CpG-free genomes to reduce TLR9 activation Tissue-specific promoters Temporary immunosuppression (steroids, rapamycin, ATG, calcineurin inhibitors) Complement inhibitors Antigen decoys or IgG-depleting strategies (plasmapheresis, IdeS) Alternative delivery routes (oral mucosa, isolated limb perfusion) Activate Treg cells and M2 macrophages
Preexisting neutralizing antibodies (Nabs)	High prevalence in human serum and synovial fluid can reduce or block transduction.	Screening patients for seroprevalence Capsid engineering for antibody escape Reduced dosing with efficient vectors Antibody depletion (plasmapheresis, IdeS) Local intra-articular immunosuppression Exploring tolerogenic delivery routes (oral mucosa, liver targeting)

## Data Availability

No new data were created or analyzed in this study. Data sharing is not applicable to this article.

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
