# Peer review of "Navigating the Hurdles of Intra-Articular AAV Gene Therapy"

_ijms, 2025, doi:10.3390/ijms262010123_

Round 1
Reviewer 1 Report
Comments and Suggestions for Authors
This manuscript systematically explores the current application status, challenges and countermeasures of AAV-mediated intra-articular gene therapy in joint diseases. The article covers the latest progress from the biological basis of AAV to preclinical and clinical trials, especially conducting in-depth analyses of key obstacles such as transduction efficiency, immunogenicity, and pre-existing immune responses. The authors also proposed a variety of engineering and immune management strategies, which have strong scientific foresight and potential for clinical transformation. Although there are some areas that can be further improved, the overall quality is relatively high, and recommending to accept it after minor revision.
- Lists multiple potential targets (such as IL-4, IL-10, TGF-β, etc.) in the "Preclinical Targets" section, but lacks in-depth discussions on the priorities, synergies or potential conflicts of these targets. Suggestingto supplement the elaboration on the logic of target selection, such as targeting strategies based on disease classification, inflammatory stage or cell type.
- Emphasizes the negative impact of immune responses on AAV treatment, but does not fully discuss how to utilize the local immune microenvironment (such as regulatory T cells and M2-type macrophages) to enhance treatment tolerance. Suggestingto supplement relevant content.
- If possible, please consider adding schematic diagrams, such as the transduction pathways of AAV within joints, immune response mechanisms, capsid engineering strategies, etc., to enhance understanding.
Author Response
We thank the reviewer for the positive evaluation of our manuscript and for the constructive suggestions. Below we provide detailed responses and describe the revisions made.
1) Preclinical targets section (IL-4, IL-10, TGF-β, etc.) lacks deeper discussion of priorities, synergies, or conflicts.
Response: We agree with this valuable suggestion. In the revised manuscript, we have expanded the “Preclinical Targets” section to discuss prioritization of targets based on disease classification (e.g., RA vs OA), inflammatory stage (acute flare vs remission), and cell types (synoviocytes, chondrocytes, macrophages):"These anti-inflammatory cytokines act in part by shifting immune cell profiles: they promote the expansion of regulatory T cell (Treg) and M2-polarized macrophages, while reducing pro-inflammatory M1 macrophages, thereby suppressing synovial inflammation and restoring immune balance. Specifically, IL-10 and IL-4
may be prioritized for conditions like RA where inflammation predominates, whereas TGF-β may be more valuable in OA, where cartilage degeneration are the main drivers (15,16)."
We also added a short discussion on potential synergies, such as the combination of multiple cytokines: "Synergistic strategies can be considered, for instance, combining antagonist of IL-1 and IL-10 delivery to simultaneously dampen inflammation and promote tolerogenic immune pathways. However, such approaches carry an increased risk of excessive immunosuppression, which could predispose patients to infection or even malignancy.""
Overall, a rational target selection requires balancing joint disease subtype, inflammatory stage, and cell-type specificity to optimize both efficacy and safety."
2) Limited discussion on harnessing the local immune microenvironment (e.g., regulatory T cells, M2 macrophages).
Response: We appreciate this point and have supplemented the manuscript with a paragraph discussing how regulatory T cells (Tregs) and M2-type macrophages can be considered to improve treatment tolerance. We now added a new paragraph as below: "
Harnessing the local immune microenvironment provides another path to decrease the immune response toward AAV. Tregs and M2-polarized macrophages are central players in maintaining immune tolerance in the joint. Expansion of Tregs can be achieved through the activation of IL-10 or TGF-β, which promotes their differentiation and suppresses effector T-cell activity. Similarly, skewing macrophages toward an M2 phenotype with cytokines such as IL-4 or IL-13 can reduce pro-inflammatory M1 phenotype and mitigate AAV immunogenicity."
3) Lack of schematic diagrams.
Response: We agree that figures can enhance clarity. In the revised manuscript, we have added a second table to clearly list the barriers and strategies respectively.
Reviewer 2 Report
Comments and Suggestions for Authors
It is a well-written paper targeting a field that AAV has great potential. Although some insights have been given regarding the future of AAV for intra-articular AAV, more highlight of major direction AAV is going in this area would be helpful. It could even be an opinion as experts in the field.
Line 114. Double check spelling of “is”
Line 135. Could you expand on tolerability issues? Did it cause pain? Major side-effects?
Section 2. Why do most of the clinical trials use AAV 5 and 2? Tropism?
Line 297. Other types of what?
Author Response
We thank the reviewer for the positive evaluation of our manuscript and for the constructive suggestions. Below we provide detailed responses and describe the revisions made.
Line 114. Double check spelling of “is.”
Response: We have carefully re-checked the spelling error and rewrote the content.
Line 135. Could you expand on tolerability issues? Did it cause pain? Major side-effects?
Response: The tolerability issues in the hIFN-β trial (ART-I02) is that several patients experienced long-lasting local adverse events, including joint swelling, periarthritis, and pain, which were considered possibly or probably related to the intra-articular injection of rAAV2/5-hIFN-β. This addition is now included in the manuscript for accuracy.
Section 2. Why do most of the clinical trials use AAV5 and AAV2? Tropism?
Response: Since only a small number of organizations have conducted clinical trials to date, with some sponsors running two or more studies, the serotype selections remain limited. Most clinical reports have focused on AAV2 and AAV5, largely because these vectors were the earliest to demonstrate efficiency in musculoskeletal tissues, and because they had already accumulated substantial preclinical experience and those serotypes transduce synovial fibroblasts and chondrocytes. However, they are not the only serotypes with potential. Studies have also shown that AAV1 and AAV6 achieve strong transduction efficiency in joint tissues (10), suggesting there may be more effective candidates for future clinical translation. We have also added this content to our manuscript.
Line 297. Other types of what?
Response: We have rewritten this paragraph.